# Effect of Different Nutritional Education Based on Healthy Eating Index for HemoDialysis Patients on Dietary Quality and Muscle Mass

**DOI:** 10.3390/nu14214617

**Published:** 2022-11-02

**Authors:** Yun-Han Chen, Wan-Lin Liu, Tuyen Van Duong, Te-Chih Wong, Hsi-Hsien Chen, Tso-Hsiao Chen, Yung-Ho Hsu, Sheng-Jeng Peng, Shwu-Huey Yang

**Affiliations:** 1School of Nutrition and Health Sciences, Taipei Medical University, Taipei City 11031, Taiwan; 2Division of Dietetics and Nutrition, Taichung Tzu Chi Hospital, Buddhist Tzu Chi Medical Foundation, Taichung City 42743, Taiwan; 3Department of Nutrition and Health Sciences, Chinese Culture University, Taipei City 11114, Taiwan; 4Division of Nephrology, Department of Internal Medicine, Taipei Medical University Hospital, Taipei City 11031, Taiwan; 5Department of Nephrology, Department of Internal Medicine, Taipei Medical University-Wan Fang Hospital, Taipei City 116081, Taiwan; 6Division of Nephrology, Department of Internal Medicine, Taipei Medical University-Shuang Ho Hospital, New Taipei City 23561, Taiwan; 7Division of Nephrology, Department of Internal Medicine, Cathay General Hospital, Taipei City 10630, Taiwan; 8Nutrition Research Center, Taipei Medical University Hospital, Taipei City 11031, Taiwan; 9Research Center of Geriatric Nutrition, College of Nutrition, Taipei Medical University, Taipei City 11031, Taiwan

**Keywords:** hemodialysis, skeletal muscle, dietitian, nutritional education, dietary quality

## Abstract

Background: Hemodialysis patients are at high risk of muscle loss as a result of aging and disease, and combined with inadequate dietary intake. The Healthy Eating Index for HemoDialysis patients (HEI-HD) was developed to assess the dietary quality of hemodialysis patients. The purposes of this study were to examine the effects of different nutritional education models using HEI-HD-based education on dietary quality and muscle mass in hemodialysis patients. Methods: A quasi-experimental study was conducted from May 2019 to April 2021, with four groups, including no course for patients and nurses (Non-C), course for nurses (CN), course for patients (CP), and course for patients and nurses (CPN). The courses were delivered by registered dietitians. The data of 94 patients were collected and analyzed at baseline, after 2 months of intervention, and 2 months follow-up, including demographics, body composition, 3-day dietary records, and hemodialysis dietary knowledge. The HEI-HD index score was calculated. Results: Patients aged 58.3 ± 10.1 years. The dietary quality change in the CPN group was improved as compared with the Non-C group (−3.4 ± 9.5 vs. 3.0 ± 5.5, 0.04). The skeletal muscle mass of the Non-C group at intervention was also significantly lower than baseline, but the CPN group was not. Conclusions: The HEI-HD-based nutritional education for both patients and nurses showed a positive effect on improving the dietary quality and maintaining muscle mass in hemodialysis patients.

## 1. Introduction

End-stage renal disease (ESRD) has been a public issue of global concern, with 90% of countries using hemodialysis (HD) as a treatment modality [1]. Most HD patients have difficulty maintaining a dietary compliance due to the stress of dietary restrictions, eventually leads to malnutrition [2]. Malnutrition in patients with chronic kidney disease is called protein-energy wasting (PEW), caused by marked reduction in storage of protein and energy in body. Muscle wasting is the most direct and valid criterion for PEW [3]. Muscle wasting is one of the direct criteria of PEW [3]. Skeletal muscle accounts for 30–40% of the whole body and maintains basal energy metabolism [4], physical activity, and daily life [5]. However, HD patients often gradually lose muscle mass with age, treatment, hormonal changes, and disease progression [6], resulting in frailty, decreased quality of life, falls, and even fractures, which increase the risk of death in the long run [7,8].

Dietary quality considers whether the individual’s diet complies with dietary recommendations and healthy diet principles [9]. The Dietary Quality Index is an index developed following national dietary guidelines or specific disease recommendations, that reflects patients’ compliance with dietary recommendations and dietary changes [10]. The Healthy Eating Index for HemoDialysis patients (HEI-HD) was developed in 2014 [11]. It was constructed based on the healthy eating pattern of HD patients [12,13,14,15,16,17,18,19,20,21]. HEI-HD contains 16 diet items. The total score of HEU-HD ranged between 0 and 100. The higher the score, the better the dietary quality, with a 4% reduction in the risk of death in HD patients [19].

The dietary principles of kidney disease are too complex to make patients have difficulty with dietary compliance [22]. Previous studies proposed that patients are reluctant to follow dietary principles because they do not have sufficient knowledge or lack of skills to self-manage their diet [23]. Therefore, appropriate education and reinforcement of knowledge concepts enhance the understanding of the HD diet principles to encourage diet compliance [24]. The Health Belief Model (HBM), as a guiding framework for educational interventions, is the most widely applied theory of health education, which describes changes in health-related behaviors depending on the personal beliefs or attitudes toward disease [25]. Shariatjafari et al. (2012) showed that the nutritional education based on HBM was effective in promoting the adherence to dietary guidelines in the participants [26].

In the routine medical care of HD centers, nurses are the most frequent contact with HD patients and often help patients to adapt to their problems by providing information [27]. Therefore, nutrition education offered by dietitians could be more helpful for HD patients. In addition, the low ratio of dietitians to patients in HD centers and nutrition care are not included in the regular care process under the health insurance system [28]. Therefore, we conducted a quasi-experimental study to examine the effects of HEI-HD-based nutritional education models on dietary quality and muscle mass in HD patients.

## 2. Materials and Methods

### 2.1. Study Design and Participants

A quasi-experimental study was conducted from May 2019 to April 2021 at HD centers of four hospitals in northern Taiwan. This study was approved by the Taipei Medical University Joint Institutional Review Board (no. N201801034) for Taipei Medical University Hospital, Taipei Medical University-Wan Fang Hospital, Taipei Medical University-Shuang Ho Hospital, and approved by the institutional Ethics Committee of Cathay General Hospital (no. CGH-OP108007) for Cathay General Hospital. All participants were fully informed and signed informed consent forms before their participation.

Patients included were those aged 20–75 years, received HD treatment thrice a week for at least 3 months, the education level of junior high school and higher, and Kt/V > 1.2. Patients with obvious edema, pregnancy, amputation, hyperthyroidism, hypothyroidism, malignancy, liver failure or cancer, mental disorders, tube feeding, hospitalization and plan to surgery, loss to measure body composition, and percentage body fat < 4% were excluded.

The study recruited 141 patients who met the criteria. All study subjects received a nutritional education booklet. The subjects were divided into four groups according to the hospital. Those were no courses for patients and nurses (Non-C) group (only provided education booklet); courses for nurses (CN) group (provided nutrition course for nurses); courses for patients (CP) group (provided nutrition course for HD patients); and courses for patients and nurses (CPN) group (provided nutrition course for nurses and HD patients). This study period was divided into baseline (T0), intervention (T1), and follow-up (T2).

### 2.2. Nutrition Education Booklet and Education Program

All participants received a nutritional education booklet. The nutritional education booklet was developed based on the HEI-HD to increase patient knowledge, promote positive attitudes, and change eating behaviors. The contents of the educational booklet were corrected by both six renal dietitians and three nephrologists. Table 1 shows the 9 chapters and the contents of the educational booklet. Nutrition education was offered at T1. A dietitian provided one-on-one, 15–20 min/week personalized nutrition education at the patient’s bedside in the first month and a 15–20 min/month personalized nutrition education with a special focus on skills to improve the low-scoring dietary items to achieve higher score in the second month. A 10-min group nutritional education session was provided by a dietitian to nurses at the beginning of T1 to address the incorrect answers to the dietary knowledge questionnaire.

### 2.3. Data Collection and Measurements

For patients at T0, socio-demographic factors, anthropometric and body composition, dietary records, and questionnaires were collected. At T1 and T2, the data collected were the same as those at T0, except for socio-demographic factors. For the nurses, only a dietary knowledge questionnaire was collected at T0, T1, and T2.

#### 2.3.1. Demographic Data

The basic information of the patients was collected by chart review, including age, gender, dialysis vintage, and comorbidity, and was calculated using the Charlson comorbidity index (CCI). In addition, the occupation, economic status, marital status, living conditions, smoking, nutrition education, and consultation status were collected using questionnaires.

#### 2.3.2. Anthropometry and Body Composition

Height and dry weight after dialysis were collected from the chart review to calculate body mass index (BMI). Skeletal muscle mass (SMM) and body fat were measured by using the bioelectrical impedance analysis (InBody S10, Biospace, Seoul, Korea) after the hemodialysis session (sitting position) [21]. The SMM was adjusted for height (SMM_Ht_^2^ (kg/m^2^)) and weight (SMM_Wt_ (%)) which were indicators of muscle mass.

#### 2.3.3. Dietary Intake

The patients recorded three days of dietary intake during the week before and after the monthly examination, including one dialysis day, one non-dialysis day, and one non-dialysis weekend day [11]. To confirm dietary records, we used 24-h dietary recall by well-trained dietitians to interview patients. Dietary intake was analyzed using nutrients analysis software (Cofit pro, Taipei, Taiwan), which is based on the 2018 Taiwan Food Nutrient Database. The components of the HEI-HD score include total grains, total protein foods, total vegetable, whole fruits, oils, high biological value proteins, the ratio of white to red meat, processed meat, fish and seafood, saturated fatty acids rich oils, sugar-sweetened beverages and fruit juice, alcohol, whole grains, nuts, taste, and milk and dairy products. Total HEI-HD score is in the range of 0–100 [19].

#### 2.3.4. Dietary Knowledge

We used Ryu’s questionnaire to assess the dietary knowledge of HD patients and nurses. It consists of 10 questions to ask patients about knowledge related to protein, potassium, phosphorus, sodium, and water. The score of the questionnaire ranged from 0 to 10; the higher the score, the better the knowledge of diet [29].

#### 2.3.5. Physical Activity

Physical activity was collected by using the short version of the International Physical Activity Questionnaire (IPAQ-SF) [30]. Patients recorded the number of days and time spent on exercise (vigorous, moderate, walking exercise, and sleep) during the last 7 days. The level of physical activity was indicated by the metabolic equivalent (MET) value.

### 2.4. Statistical Analysis

Data were shown as mean with standard deviation (SD) for continuous variables and number (*n*) with percentage (%) for categorical variables. A Shapiro–Wilk test or Kolmogorov–Smirnov test was used to verify the normal distribution assumption for continuous variables. One-way ANOVA with post hoc Bonferroni’s test was used to compare demographics, changes in the dietary knowledge, and HEI-HD. Changes in the skeletal muscle mass was compared using the ANCOVA by adjusting for gender and age, followed by Bonferroni’s test. Kruskal–Wallis test was used to compare non-normal distribution data between multiple groups. The Chi-square test was used to compare categorical variables. Paired *t*-test was used to compare continuous variables within groups. All statistical analyses were used by SAS version 9.4. A *p*-value < 0.05 was considered statistical significance, and 0.05 < *p* < 0.1 indicated marginal significant differences.

## 3. Results

### 3.1. Characteristics of Participants

Ninety-four HD patients’ data were analyzed (Figure 1). The mean age of the HD patients was 58.3 ± 10.1 years, 64.9% were men. The means of total HEI-HD score, SMM were 65.6 ± 8.3, 26.3 ± 5.7 kg, respectively. There was a statistically significant difference of marital status existed among the four groups (Table 2).

### 3.2. Comparison of Changes in Dietary Knowledge among the Groups

The dietary knowledge score in the Non-C group at T2 was significantly higher than at T1 (Table 3).

### 3.3. Comparison of Changes in HEI-HD among the Groups

After two months of intervention, there was a significant increase in vegetable scores in the CPN group and the CP group but not in the Non-C group. The ratio of white to red meat score changes in the CPN group was significantly increased. In comparison with T0, at the end of T2, the vegetable scores change in the CP group were significantly higher than the Non-C group, the ratio of white to red meat scores of the CPN group was significantly higher than the CP group, and the total HEI-HD scores of the CPN group was significantly higher than the Non-C group. There was no significant difference among the four groups in T2 to T1 period (Table 4).

### 3.4. Comparison of Changes in Skeletal Muscle Mass among the Groups

There was no significant difference in muscle mass among the four groups. To compare with T0, all patients’ SMM, SMM_Ht_^2^, and SMM_Wt_ were significantly decreased at T1. The SMM and SMM_Ht_^2^ of the Non-C group at T1 were also significantly lower than T0. At T2, all patients’ SMM, SMM_Ht_^2^, and SMM_Wt_ were significantly higher than T1, but only SMM of the Non-C group and the CN group were significantly better than T1 with or without adjusted (Table 5).

## 4. Discussion

This was the first study to assess the effects of dietitians providing nutritional education for nurses and patients alone and in combination based on HEI-HD on dietary quality and muscle mass of HD patients. This study showed that nutritional education for the CPN group, both patients and nurses by dietitians, based on HEI-HD was more effective than other groups. The dietitians provided nutritional education and increased the scores of vegetable, the ratio of white to red meat, and total HEI-HD, but no significant change in SMM, SMM_Ht_^2^, and SMM_Wt_.

Ford et al. (2004) showed that giving nutrition education for 20–30 min per month to HD patients could improve patients’ dietary knowledge [24]. Abd et al. (2015) used small group combined educational booklets for nutrition education for the study groups. Their results reported dietary knowledge would be improved [31]. In this present study, the dietary knowledge of HD patients improved at T2, which compared to T0, the Non-C group had a higher dietary knowledge score at T2 than T1, indicating that only providing the HEI-HD booklet could help patients’ dietary knowledge improvement.

Vegetables are rich in vitamins C and E and antioxidant phytochemicals [32]. Saglimbene et al. (2019) reported that only 4% HD patients achieved vegetable recommendations, which was far below the recommendations for preventing chronic diseases [33]. Wagner et al. (2016) showed that nutrition courses given by dietitians could enhance the vegetable intake of obese adults [34], which was similar to the results of our study, indicating that nutrition education by dietitians could increase the vegetable intake of patients and increase the dietary vegetable scores.

The ratio of white to red meat scores improved. Increasing white meat intake daily could reduce 33% mortality [19]. HD patients often have excessive red meat consumption problems. Red meat, such as pork, beef, and lamb, which are rich in phosphorus and saturated fat, could increase the risk of cardiovascular disease [35]. McCullough et al. (2002) recommended that the ratio of white and red meat is 4:1 to prevent cardiovascular disease [36]. The ratio of white to red meat scores of the CPN group was better than the others in this study, which means that the nutrition education given to both patients and nurses by dietitians could significantly increase HD patients’ dietary ratio of white and red meat.

This present study showed that the total HEI-HD scores of the CPN group were significantly higher than others due to the increase in vegetable scores and the ratio of white to red meat scores. Previous studies have shown that people who have received nutrition education have better dietary quality. Nutrition education by dietitians could improve the dietary quality of diabetes patients [37,38]. Our study shows that nutritional education by dietitians to patients and nurses had the best effect on improving the dietary quality of HD patients.

Previous studies have shown that increased dietary quality could maintain muscle mass [39,40]. Rondanelli et al. (2015) reported that the intake of white meat increased combined with the intake of red meat maintained or decreased, which was able to prevent the occurrence of sarcopenia [41], similar to our study in which no significant decrease in muscle mass was observed in the CPN group with enhanced dietary quality. In contrast, the total HEI-HD score of the Non-C group was slightly lower, and their SMM and SMM_Ht_^2^ were significantly reduced at T1. In addition, previous studies showed that higher educated people are more willing to exercise [42], and unmarried people and blue-collar workers also have high physical activity [43]. Our study results showed that the higher education level in the Non-C group and more unmarried, more blue-collar workers in the CN group tend to have relatively high levels of activity, which might affect muscle mass. Our study’s strengths: (1) this was a multi-center study, which could reduce the sampling bias compared to a single center; (2) this was the first nutrition education study that utilizes four educational models; and (3) this was the first nutrition education study using the HEI-HD, a simple and validated dietary index scale that could quickly assess patients’ dietary status. There are some limitations in this study as follows: (1) only 4 months of the study were monitored, which is relatively short and it may not be possible to see the increase in muscle mass; (2) the effects of diet on muscle mass were limited, so physical activities, exercise and nutrition supplementation should be considered to be included in the future study; (3) the subjects were all voluntary participants with high health awareness, so the nutrition education may be more effective; (4) no adequate information was provided regarding the frequency and content of nurse education for patient [44], so future studies should monitor the frequency and how long for nurse education; and (5) it should be noted that the present study is a multi-center study and was conducted only in northern Taiwan urban with high literacy population. Therefore, the results obtained may not be appropriate to apply to patients in Taiwan rural areas or other countries.

## 5. Conclusions

The HEI-HD-based nutritional education was effective in improving patients’ dietary quality, and the dietitians’ model educating both patients and nurses was the most effective in improving dietary quality and maintaining skeletal muscle mass in HD patients.

## Figures and Tables

**Figure 1 nutrients-14-04617-f001:**
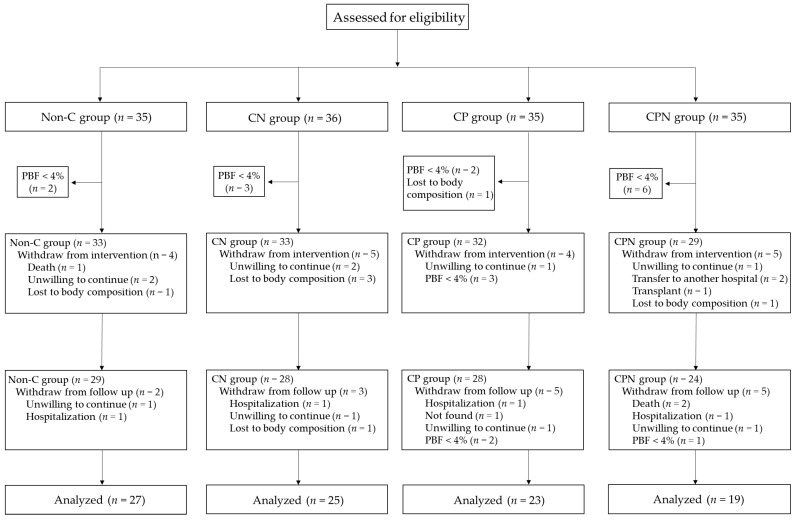
Study flow of HD patients. Non-C, no course for patients and nurses; CN, course for nurses; CP, course for patients; CPN, course for patients and nurses; PBF, percentage body fat.

**Table 1 nutrients-14-04617-t001:** The chapters and the contents of the education booklet.

Chapter	Content
Chapter 1	Recommendations for Dietary Requirements
Chapter 2	Definition of serving
Chapter 3	Healthy diet for dialysis patients Enough energy intake Enough protein intake Healthy diet intake Control dietary intake of phosphorus, potassium, and sodium Iron food intake Fluid intake restriction Dairy and dairy products are not recommended for consumption
Chapter 4	Tips for food intake
Chapter 5	Food label information
Chapter 6	Instructions for phosphorus-binding drugs
Chapter 7	Exercise rules
Chapter 8	Body weight record form
Chapter 9	Diet record sheet and HEI-HD calculated table

HEI-HD: Healthy Eating Index for HemoDialysis patients.

**Table 2 nutrients-14-04617-t002:** Baseline characteristics of all participants.

	All (*n* = 94)	Non-C (*n* = 27)	CN (*n* = 25)	CP (*n* = 23)	CPN (*n* = 19)	*p* Value
Age, years	58.3 ± 10.1	60.1 ± 11.1	54.5 ± 10.7	60.3 ± 7.1	58.5 ± 10.2	0.14
Male, *n* (%)	61 (64.9)	17 (48.6)	16 (64.0)	18 (78.3)	10 (52.6)	0.37
post-HD weight, kg	64.5 ± 14.3	64.5 ± 17.6	63.6 ± 10.1	69.6 ± 13.8	59.3 ± 13.2	0.13
Height, cm	164.4 ± 8.5	164.7 ± 10.3	163.6 ± 6.5	167.3 ± 7.9	162.5 ± 8.3	0.28
BMI, kg/m^2^	23.6 ± 4.0	23.5 ± 4.4	23.8 ± 3.4	24.8 ± 4.6	22.2 ± 3.3	0.23
HD vintage, y	5.8 ± 5.4	7.5 ± 6.8	5.1 ± 3.9	5.9 ± 6.2	4.3 ± 3.4	0.22
MET, kcal/day, median (tertile1, tertile3)	594 (271.4, 1386)	808.5 (411.8, 2772)	426 (122.2, 1319.5)	426 (279.7, 1109.3)	547.5 (267.3, 1475.5)	0.51
CCI	2.8 ± 0.8	3.0 ± 0.9	2.7 ± 0.8	2.8 ± 0.7	2.5 ± 0.8	0.25
SMM, kg	26.3 ± 5.7	25.2 ± 7.1	27.0 ± 4.0	28.2 ± 4.6	24.7 ± 6.3	0.15
Total HEI-HD score	65.6 ± 8.3	67.3 ± 9.1	63.3 ± 7.5	67.0 ± 9.4	64.6 ± 6.5	0.25
Dietary knowledge scores	7.1 ± 1.8	7.2 ± 1.7	7.5 ± 1.6	6.8 ± 1.9	6.7 ± 1.9	0.39
Education level						0.33
Junior, *n* (%)	19 (20.9)	2 (8.0)	3 (12.5)	9 (39.1)	5 (26.3)	
Senior, *n* (%)	38 (41.8)	12 (48.0)	11 (45.8)	8 (34.8)	7 (36.8)	
Colleges, *n* (%)	29 (31.9)	10 (40.0)	9 (37.5)	5 (21.7)	5 (26.3)	
≥Master, *n* (%)	5 (5.5)	1 (4)	1 (4.2)	1 (4.4)	2 (10.5)	
Occupation						0.53
Public employees, *n* (%)	9 (9.6)	2 (7.4)	2 (8.0)	2 (8.7)	3 (15.8)	
Service industry, *n* (%)	8 (8.5)	4 (14.8)	3 (12.0)	0 (0)	1 (5.3)	
Industrial and commercial, *n* (%)	9 (9.6)	0 (0)	5 (20.0)	3 (13.0)	1 (5.3)	
Freelance, *n* (%)	12 (12.8)	3 (11.1)	2 (8.0)	4 (17.4)	3 (15.8)	
Retirement, *n* (%)	36 (38.3)	13 (48.1)	7 (28.0)	9 (39.1)	7 (36.8)	
Unemployed, *n* (%)	4 (4.3)	0 (0)	1 (4.0)	1 (4.4)	2 (10.5)	
Others, *n* (%)	16 (17.0)	5 (18.5)	5 (20.0)	4 (17.4)	2 (10.5)	
Marital status						0.03
Single, *n* (%)	18 (20.9)	3 (12.5)	9 (39.1)	4 (20.0)	2 (10.5)	
Married, *n* (%)	56 (65.1)	19 (79.2)	9 (39.1)	14 (70.0)	14 (73.7)	
Divorced/Widowed, *n* (%)	12 (14.0)	2 (4.0)	5 (21.7)	2 (10.0)	3 (15.8)	
Live alone, *n* (%)	7 (5.0)	2 (8.3)	3 (8.3)	1 (2.9)	1 (2.9)	0.66
Accepted nutritional education/counseling before						0.43
Yes, *n* (%)	74 (79.6)	22 (62.9)	18 (72.0)	17 (73.9)	17 (89.5)	
Nurses						0.20
Age, years	37.7 ± 7.7	39.1 ± 9.7	35.3 ± 4.6	38.0 ± 7.8	38.4 ± 7.9	
Dietary knowledge scores	8.1 ± 1.1	8.2 ± 1.2	8.0 ± 1.0	8.0 ± 1.1	8.2 ± 0.9	0.79

HEI-HD, The Healthy Eating Index in HemoDialysis patients; SMM, skeletal muscle mass; Non-C, no course for patients and nurses; CN, course for nurses; CP, course for patients; CPN, course for patients and nurses; HD, hemodialysis; BMI, body mass index; CCI, Charlson comorbidity index. Data were presented as mean ± SD, number (percentage), or median (tertile1, tertile3) (*n* = 94). *p* value obtained from ANOVA test (continuous variables), Kruskal–Wallis test (non-normal distribution continuous variables), or Chi-squared tests (categorical variables).

**Table 3 nutrients-14-04617-t003:** Changes in the total scores of dietary knowledge questionnaire in different nutritional education models.

	All (*n* = 94)	Non-C (*n* = 27)	CN *(n* = 25)	CP (*n* = 23)	CPN (*n* = 19)	*p* Value
T1-T0	0.1 ± 1.6	−0.3 ± 1.8	0.3 ± 1.6	0.3 ± 1.3	0.3 ± 1.6	0.48
T2-T0	0.4 ± 1.6 *	0.4 ± 1.8	0.3 ± 1.6	0.4 ± 1.7	0.3± 1.5	0.99
T2-T1	0.2 ± 1.5	0.7 ± 1.4 *	0.0 ± 1.3	0.2 ± 1.6	0.0 ± 1.7	0.35

Non-C, no course for patients and nurses; CN, course for nurses; CP, course for patients; CPN, course for patients and nurses. Data were presented as mean ± SD (*n* = 94). T0, baseline; T1, intervention; T2, follow up. * *p* < 0.05 indicates significant differences within the group using paired *t*-test.

**Table 4 nutrients-14-04617-t004:** Changes in the HEI-HD and its 16 items score in different nutritional education models.

	All (*n* = 94)	Non-C (*n* = 27)	CN (*n* = 25)	CP (*n* = 23)	CPN (*n* = 19)	*p* Value ^1^
T1-T0						
Total grains	−0.2 ± 2.1	−0.4 ± 2.1	−0.3 ± 1.8	0.4 ± 2.4	−0.3 ± 1.8	0.48
Total protein foods	−0.6 ± 2.1	−0.3 ± 2.2	−1.2 ± 2.1 *	0.1 ± 2.0	−1.0 ±1.9 *	0.12
Total vegetable	−0.5 ± 2.5	−1.7 ± 2.6 ^c^*	−0.8 ± 2.7 ^abc^	0.2 ± 1.5 ^ab^	0.5 ± 2.7 ^a^	<0.01
Whole fruits	−0.3 ± 1.5 *	−0.2 ± 1.7	−0.1 ± 1.2	−0.5 ± 1.9	−0.4 ± 1.1	0.80
Oils	−0.3 ± 1.7	−0.2 ± 1.4	−0.4 ± 1.7	−0.4 ± 2.1	−0.3 ± 1.5	0.95
HBV proteins	0.1 ± 0.4	0.1 ± 0.3	0.2 ± 0.6 ^#^	0.1 ± 0.5	−0.01 ± 0.1	0.27
The ratio of white to red meat	−0.6 ± 2.1 *	−1.2 ± 2.3 ^b^*	−0.3 ± 1.6 ^ab^	−1.1 ± 2.0 ^ab^*	0.5 ± 2.0 ^a^	0.03
Processed meat	0.3 ± 2.0	0.5 ± 2.4	0.2 ± 1.9	−0.3 ± 1.8	0.9 ± 1.7 *	0.26
Fish and seafood	0.3 ± 2.7	0.5 ± 2.5	0.6 ± 2.9	0.1 ± 3.1	−0.3± 2.5	0.64
SFA-rich oils	−0.02 ± 1.2	−0.1 ± 1.1	−0.5 ± 0.8 *	0.2 ± 1.4	0.4 ± 1.7	0.09
Sugar-sweetened beverages and fruit juice	0.0 ± 1.8	0.2 ± 1.7	0.2 ± 2.0	−0.7 ± 1.6 *	0.3 ± 2.0	0.24
Alcohol	−0.1 ± 0.8	−0.1 ± 1.1	0.0 ± 0.0	−0.1 ± 0.3	−0.1 ± 1.3	0.99
Whole grains	−0.1 ± 0.9	−0.2 ± 0.5 ^#^	0.1 ± 1.1	−0.1 ± 1.2	−0.1 ± 0.4	0.58
Nuts and seeds	−0.1 ± 1.2	−0.2 ± 1.3	0.0 ± 1.3	−0.1 ± 1.3 ^#^	0.2 ± 1.0	0.32
Taste	0.05 ± 1.0	−0.1 ± 0.8	0.1 ± 1.4	−0.1 ± 0.9	0.4 ± 1.1	0.31
Milk and dairy products	0.04 ± 2.0	0.2 ± 2.2	−0.5 ± 2.2	−0.2 ± 1.7	0.7 ± 1.6 ^#^	0.23
Total HEI-HD score	−2.3 ± 8.4	−3.3 ± 9.1 ^#^	−3.2 ± 9.7	−2.7 ± 7.3 ^#^	0.8 ± 6.4	0.36
T2-T0						
Total grains	−0.03 ± 1.9	−0.4 ± 2.0	−0.1 ± 1.6	0.3 ± 1.8	0.1 ± 2.2	0.78
Total protein foods	−0.1 ± 1.9	0.1 ± 2.5	−0.5 ± 1.4	0.3 ± 1.6	−0.4 ± 1.8	0.44
Total vegetable	−0.3 ± 2.8	−1.6 ± 2.4 ^b^*	−0.3 ± 2.6 ^ab^	0.7 ± 2.7 ^a^	0.1 ± 3.4 ^ab^	0.03
Whole fruits	−0.3 ± 1.6	−0.2 ± 1.6	−0.3 ± 1.8	−0.4 ± 1.5	−0.3 ± 1.6	0.95
Oils	−0.1 ± 1.4	−0.1 ± 1.3	−0.1 ± 1.1	−0.3 ± 1.5	0.3 ± 1.7	0.66
HBV proteins	0.1 ± 0.5	0.1 ± 0.3	0.2 ± 0.7	0.1 ± 0.5	0.0 ± 0.0	0.52
The ratio of white to red meat	−0.6 ± 2.0	−1.1 ± 2.3 ^ab^*	−0.3 ± 1.7 ^ab^	−1.3 ± 2.0 ^b^*	0.4 ± 1.8 ^a^	0.03
Processed meat	−0.01 ± 2.2	0.1 ± 2.7	0.1 ± 2.4	−0.4 ± 1.7	0.1 ± 1.6	0.81
Fish and seafood	0.3 ± 2.7	0.1 ± 2.7	0.5 ± 3.0	0.2 ± 2.6	0.6 ± 2.4	0.94
SFA-rich oils	−0.2 ± 1.1	−0.4 ± 1.2	−0.5 ± 0.9 *	−0.1 ± 1.1	0.3 ± 1.1	0.12
Sugar-sweetened beverages and fruit juice	0.2 ± 2.4	1.1 ± 2.0	0.1 ± 2.1	−0.03 ± 2.6	1.1 ± 2.0 *	0.27
Alcohol	0.04 ± 0.8	−0.1 ± 1.0	−0.1 ± 0.3	0.0 ± 0.0	0.4 ± 1.3	0.26
Whole grains	0.0 ± 0.7	−0.1 ± 0.3	0.1 ± 1.1	0.0 ± 0.7	−0.1 ± 0.4	0.70
Nuts and seeds	−0.2 ± 1.3	−0.2 ± 1.4	−0.1 ± 1.5	−0.4 ± 1.0	−0.3 ± 1.4	0.94
Taste	0.2 ± 1.5	0.4 ± 2.0	0.1 ± 1.0	0.0 ± 1.6	0.1 ± 1.2	0.79
Milk and dairy products	−0.2 ± 2.0	−0.1 ± 2.2	−0.3 ± 2.0	−0.8 ± 1.8 *	0.6 ± 1.8	0.13
Total HEI-HD score	−1.3 ± 7.8	−3.4 ± 9.5 ^b#^	−1.7 ± 7.4 ^ab^	−2.0 ± 6.7 ^ab^	3.0 ± 5.5 ^a^*	0.05
T2-T1						
Total grains	0.2 ± 1.7	0.1 ± 1.7	0.2 ± 1.9	−0.1 ± 1.6	0.5 ± 1.4	0.75
Total protein foods	0.5 ± 1.7	0.4 ± 1.9	0.7 ± 1.9 ^#^	0.2 ± 1.7	0.5 ± 1.4	0.72
Total vegetable	0.2 ± 2.5	0.1 ± 2.4	0.5 ± 2.7	0.5 ± 2.3	−0.4 ± 2.6	0.62
Whole fruits	0.02 ± 1.2	0.03 ± 1.1	−0.1 ± 1.4	0.1± 1.2	0.1 ± 1.2	0.88
Oils	0.2 ± 1.5	0.02 ± 1.7	0.3 ± 1.5	0.1 ± 1.3	0.5 ± 1.5	0.72
HBV proteins	0.0 ± 0.3	0.0 ± 0.0	−0.02 ± 0.5	0.01 ± 0.1	0.01 ± 0.1	0.97
The ratio of white to red meat	−0.03 ± 1.8	0.1 ± 2.1	−0.01 ± 1.4	−0.2± 1.8	−0.1 ± 1.9	0.95
Processed meat	−0.3 ± 2.0	−0.4 ± 2.4	−0.1 ± 1.9	−0.1± 2.0	−0.8 ± 1.4 *	0.61
Fish and seafood	0.07 ± 2.6	−0.4 ± 1.9	−0.1 ± 2.7	0.1 ± 2.8	0.9 ± 2.7	0.35
SFA-rich oils	−0.2 ± 1.2	−0.3 ± 1.1	0.001 ± 1.4	−0.3 ± 1.2	−0.1 ± 1.3	0.78
Sugar-sweetened beverages and fruit juice	0.2 ± 2.3	−0.4 ± 0.2	−0.1 ± 1.9	0.7 ± 2.7	0.9 ± 2.4	0.19
Alcohol	0.1 ± 0.8	0.0 ± 0.9	−0.1 ± 0.3	0.1 ± 0.3	0.4 ± 1.3	0.20
Whole grains	0.1 ± 0.6	0.1 ± 0.6	0.0 ± 0.5	0.1 ± 0.7	0.0 ± 0.6	0.76
Nuts and seeds	−0.1 ± 1.3	−0.03 ± 1.0	−0.1 ± 1.3	0.1 ± 1.6	−0.4 ± 1.2	0.52
Taste	0.1 ± 1.6	0.6 ± 1.8	0.1 ± 1.7	0.1 ± 1.6	−0.3 ± 1.2	0.32
Milk and dairy products	−0.2 ± 1.8	−0.4 ± 1.9	0.2 ± 1.9	−0.7 ± 2.1	−0.1 ± 1.4	0.42
Total HEI-HD score	1.0 ± 7.9	−0.1 ± 8.6	1.5 ± 10.0	0.7 ± 6.1	2.3 ± 6.0	0.78

HEI-HD, The Healthy Eating Index in Hemodialysis patients; HBV, high biological value; SFA, saturated fatty acids; non-C, no course for patients and nurses; CN, course for nurses; CP, course for patients; CPN, course for patients and nurses; T0, baseline; T1, intervention; T2, follow-up. Data are presented as mean ± SD. ^1^ Different superscripts a, b, c denote significant differences between paired groups a and b by ANOVA, followed by Bonferroni’s test (*p* < 0.05). * *p* < 0.05 indicates significant differences within the group using the paired-*t* test, and ^#^ 0.1 < *p* < 0.05 indicates significant marginal differences.

**Table 5 nutrients-14-04617-t005:** Changes in the muscle mass in different nutritional education models.

	All (*n* = 94)	Non-C (*n* = 27)	CN (*n* = 25)	CP (*n* = 23)	CPN (*n* = 19)	*p* Value
T1-T0						
SMM, kg	−0.7 ± 2.1 *	−0.9 ± 1.8 *	−0.3 ± 1.7	−0.5 ± 1.9	−1.3 ± 3.1	0.41
SMM_Ht_^2^, kg/m^2^	−0.3 ± 0.7 *	−0.3 ± 0.6 *	−0.1 ± 0.6	−0.2 ± 0.7	−0.5 ± 1.1	0.51
SMM_Wt_, %	−0.9 ± 3.5 *	−0.8 ± 3.0	−0.7 ± 2.6	−0.3 ± 3.4	−2.1 ± 5.1	0.83
T2-T0						
SMM, kg	−0.1 ± 2.9	−0.2 ± 1.8	0.5 ± 2.3	0.3 ± 3.8	−1.3 ± 3.4	0.84
SMM_Ht_^2^, kg/m^2^	−0.0 ± 1.0	−0.0 ± 0.6	0.2 ± 0.9	0.1 ± 1.3	−0.4 ± 1.2	0.87
SMM_Wt_, %	0.1 ± 3.8	0.2 ± 2.4	0.5 ± 3.6	0.3 ± 4.6	−1.0 ± 4.7	0.53
T2-T1						
SMM, kg	0.6 ± 2.2 *	0.6 ± 1.1 *	0.8 ± 1.2 *	0.8 ± 3.4	0.0 ± 2.7	0.78
SMM_Ht_^2^, kg/m^2^	0.2 ± 0.8 *	0.2 ± 0.4 *	0.3 ± 0.5 *	0.3 ± 1.1	0.0 ± 1.0	0.78
SMM_Wt_, %	1.0 ± 2.7 *	1.0 ± 1.4 *	1.2 ± 1.9 *	0.6 ± 3.9	1.0 ± 3.2	0.46

Non-C, no course for patients and nurses; CN, course for nurses; CP, course for patients; CPN, course for patients and nurses; SMM, skeletal muscle mass; SMM_Ht_^2^, skeletal muscle mass adjusted for body height; SMMwt, skeletal muscle mass adjusted for body weight; T0, baseline, T1, intervention, T2, follow up. * *p* < 0.05 indicates significant differences within the group using the paired *t*-test.

## Data Availability

The data presented in this study are available on request from the corresponding author. The data are not publicly available due to the Taipei Medical University-Joint Institutional Review Board privacy protection policy.

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
