# Peer review of "Effect of Different Nutritional Education Based on Healthy Eating Index for HemoDialysis Patients on Dietary Quality and Muscle Mass"

_nutrients, 2022, doi:10.3390/nu14214617_

Round 1
Reviewer 1 Report
The present manuscript lacks clarity.
1) Why does a study conduct from May 2019 to April 2021 lead to patient monitoring of only 4 months?
2) What is the nurses' approach towards patients?
3) If we are talking about fluid restriction it would be advisable to conduct a BIA test even before the dialysis session.
4) It is unclear how changes in patients' diets occur (one diet day on dialysis day, one diet day on a day without dialysis, and one weekend day).
5) Four months is a short time to evaluate the changes in lean body mass
6) According to the HEI-HD it has been suggested: no consumption of processed meat and a consuption of less than 1.5 portions of red meat per day to obtain the maximum score. What is meant by portion? It would be appropriate to quantify it. In any case, the consumption of red meat is excessive (see Okinawan diet)
Author Response
Dear Reviewer:
We really appreciate your advisement, it so helpful to improve our article.
1) Why does a study conduct from May 2019 to April 2021 lead to patient monitoring of only 4 months?
Ans: Thank you for your comment, we will discuss it in limitation.
2) What is the nurses' approach towards patients?
Ans: Shahdadi & Rahnama (2018) proposed that in the routine medical care of HD centers, nurses are the most frequent contact with HD patients and often help patients to adapt to their problems by providing information, introduction 4th section.
3) If we are talking about fluid restriction it would be advisable to conduct a BIA test even before the dialysis session.
Ans: Thank you for your comment, we take BIA after dialysis session that because we try to get lean body mass data and so on, not for the purpose of fluid restriction.
4) It is unclear how changes in patients' diets occur (one diet day on dialysis day, one diet day on a day without dialysis, and one weekend day).
Ans: Table 2, total HEI-HD score and table 4 the changes of the HEI-HD expressed as dietary change during the experimental period. HD patients recorded 3 days of dietary intake during the week before and after the monthly examination, including one dialysis day, one non-dialysis day, and one non-dialysis weekend day.
5) Four months is a short time to evaluate the changes in lean body mass
Ans: Thank you for your comment, we will discuss it in limitation
6) According to the HEI-HD it has been suggested: no consumption of processed meat and a consumption of less than 1.5 portions of red meat per day to obtain the maximum score. What is meant by portion? It would be appropriate to quantify it. In any case, the consumption of red meat is excessive (see Okinawan diet)
Ans: Thank you for your comment, we are going to change “portion” to “serving”.
Reviewer 2 Report
This study was not only focused on dietary knowledge, but also adherence/behavioral change by focusing on nutrition education. It is challenging to anticipate muscle mass may be different through an educational program that did not highlight proteins significantly.
Title: Recommend using different terminology than nutrition educational models as this study focused on proving the information with different groups, not necessarily using different models to educate someone.
Abstract: Generally dietitians do not educate based on healthy eating index as this is a method to obtain how well or not someone is adhering to the Dietary Guidelines for Americans. Instead dietitians educate based on recommendations for a specific disease state, therefore clarify this information. From the methods and the background, the argument is that HD patients lose muscle mass. As body composition was collected, were any differences seen with this during this intervention? Please provide this critical information or suggest revising the argument about muscle mass and instead focusing on diet intake of those on HD tend to be inadequate.
Introduction:
In the first paragraph, disconnect between the discussion about HD and complications associated with it and diet. May consider first introducing that diet is a potential factor to these complications….Then, suggest discussing the general recommendations for a dialysis patient and how diet quality helps to determine if a dialysis patient is consuming close to the recommendations and where education may be needed. Compared to the HEI-2015 used in the US, indicate what is different in the HD-HEI as the HEI-2015 contains 13 components with overall diet quality.
Considering that this study was conducted in Taiwan, would consider including nutrition education programs specific to dialysis patients in different countries and the effectiveness to them to demonstrate why this was preferred to potential other methods to employ with patients.
Methods:
Expand on why those with a certain education level could only participate. Include references to why individuals with <4% body fat was excluded. As this study was also looking at muscle mass differences, were there any inclusion criterion for this? If so, include. If someone had eating disorders or restrictive diets, were they excluded or included? Please include.
For the nutrition education booklet, please include which types of experts (renal dietitians, nephrologists, etc). Include if this booklet was in a language primarily spoken by the participants. Please include the total length of the intervention.
Lines 101-103 in the methods, the way this is written indicates that a dietitian may have tailored information in order for a patient to achieve a higher diet quality score, which would not be the intention of this nutrition education program. Please revise this portion.
For the intervention, how was the nutrition education provided? Meaning was it just going through the chapters each week? How was 15-20 minutes determined as ideal?
Expand on the bioelectrical impedance as far as if the patient stood or how it was collected and reference about collecting post dialysis. Also, indicate if the patient could not have drank anything prior to measurement.
For the dietary recall, that one weekend day may have been a dialysis day or non, correct? Generally it is 2 dialysis days and 1 non-dialysis day or 2 non-dialysis days and 1 dialysis day for dietary collection. Meaning weekend days and generally not considered different than dialysis or non-dialysis days. Even though the HD-HEI was described in another paper, please indicate what was the range of scores (e.g., 0-50)
Results:
As it is unknown the scoring of the nutrition knowledge questionnaire, it is not known if these scores were ‘good’ or not. Recommend including the scoring of this questionnaire in the methods.
Discussion:
Considering that HD patients may be restricted to consuming certain foods due to serum levels, may consider including that information as the reason some changes were not observed regardless of the education group that they were in. As eating more vegetables and less red meat may be beneficial in the long-run, it is difficult to make this conclusion as no serum was collected and it is not known the impact this will have on HD patients.
For the muscle mass, there is very little data to suggest that education alone could improve muscle status in HD patients as this was a short-term study. Even though diet quality is the outcome of this study, understanding the total amount of protein the participants were consuming and how this may have been the reason limited differences were seen could have been mentioned. Also, it is not known the actual labs of these individuals to see if there were improvements in protein status, not only in muscle mass. Important to mention.
Finally, it appears no educational theory was used to guide this study. It may have been a way to help explain some of the findings.
Author Response
Please see the attachement

Reviewer 3 Report
This is an interesting manuscript that indicated that HEI-HD-based nutritional has a positive effect on improving the dietary quality and muscle mass in hemodialysis patients. Nevertheless, I have some comments:
1) I recommend to the authors to apply Benjamini-Hochberg’s correction or Bonferroni’s correction to adjust the p-values for for multiple comparison testing in all the tables and figures.
2) Muscle mass depend on and gender. So, I suggest performing an adjusted model by age, IMC and gender (p.e. ANCOVA) in order to study changes in muscle mass in the different nutritional education model groups (table 5).
3) In conclusion, in limitation section, I would include a statement about the limited sample size of each group and its repercussion. And another statement about that this study is unicenter and its results are not generalizable to other population.
Author Response
Dear Reviewer:
We really appreciate your advisement, it so helpful to improve our article.
This is an interesting manuscript that indicated that HEI-HD-based nutritional has a positive effect on improving the dietary quality and muscle mass in hemodialysis patients. Nevertheless, I have some comments:
1) I recommend to the authors to apply Benjamini-Hochberg’s correction or Bonferroni’s correction to adjust the p-values for for multiple comparison testing in all the tables and figures.
Ans: Thank you for your comment, we have changed the Bonferroni’s correction instead of Tukey’s.
2) Muscle mass depend on and gender. So, I suggest performing an adjusted model by age, IMC and gender (p.e. ANCOVA) in order to study changes in muscle mass in the different nutritional education model groups (table 5).
Ans: Thank you for your comment, we have changed the ANCOVA instead of ANOVA.
3) In conclusion, in limitation section, I would include a statement about the limited sample size of each group and its repercussion. And another statement about that this study is unicenter and its results are not generalizable to other population.
Ans: Thank you for your comment, we have modified it
Round 2
Reviewer 2 Report
The authors have revised the manuscript significantly. However, in the methods section, lines 141-142 please signify the number of dietitians/ nephrologists who corrected the content in the booklet and if this was piloted among individuals prior to actual intervention. Furthermore, as the revised sections had some grammatical/technical writing errors, would suggest an external individual review to ensure the reading/interpretation is accurate.
Author Response
Dear reviewer,
Thank you so much for your opinion.
141-142 please signify the number of dietitians/ nephrologists who corrected the content in the booklet and if this was piloted among individuals prior to actual intervention.
Ans: There are 3 nephrologists, and 6 dietitians help us correct the booklet. We spread the booklet after corrected completely.
Furthermore, as the revised sections had some grammatical/technical writing errors, would suggest an external individual review to ensure the reading/interpretation is accurate.
Ans: Thank you so much for your opinion.
Reviewer 3 Report
The authors have performed all the suggested corrections. In my opinion, it could be accepted in the present form.
Author Response
Dear reviewer,
Thank you so much.